# 0.5 V Multiple-Input Fully Differential Operational Transconductance Amplifier and Its Application to a Fifth-Order Chebyshev Low-Pass Filter for Bio-Signal Processing

**DOI:** 10.3390/s24072150

**Published:** 2024-03-27

**Authors:** Tomasz Kulej, Fabian Khateb, Montree Kumngern

**Affiliations:** 1Department of Electrical Engineering, Czestochowa University of Technology, 42-201 Czestochowa, Poland; kulej@el.pcz.czest.pl; 2Department of Microelectronics, Brno University of Technology, Technická 10, 601 90 Brno, Czech Republic; khateb@vutbr.cz; 3Faculty of Biomedical Engineering, Czech Technical University in Prague, nám. Sítná 3105, 272 01 Kladno, Czech Republic; 4Department of Electrical Engineering, Brno University of Defence, Kounicova 65, 662 10 Brno, Czech Republic; 5Department of Telecommunications Engineering, School of Engineering, King Mongkut’s Institute of Technology Ladkrabang, Bangkok 10520, Thailand

**Keywords:** bulk-driven, multiple input MOS transistor, low-pass filter, low-voltage low-power CMOS

## Abstract

This paper presents a multiple-input fully differential operational transconductance amplifier (MI-FD OTA) with very low power consumption. To obtain a differential MOS pair with minimum supply voltage and minimum power consumption, the multiple-input bulk-driven MOS transistor operating in the subthreshold region is used. To show the advantage of the MI-FD OTA, a fifth-order Chebyshev filter was used to realize a low-pass filter capable of operating with a supply voltage of 0.5 V and consuming 60 nW at a nominal setup current of 3 nA. The proposed filter uses five MI-FD OTAs and five capacitors. The total harmonic distortion (THD) was 0.97% for a rail-to-rail sinusoidal input signal. The MI-FD OTA and the filter application were designed and simulated in the Cadence environment using a 0.18 µm CMOS process from TSMC. The robustness of the design was confirmed by Monte Carlo analysis and process, voltage, and temperature corner analysis.

## 1. Introduction

The operational transconductance amplifier (OTA) is a fundamental building block in analog signal processing. It has been used in a wide variety of applications, such as analog filters, audio amplifiers, data converters (D/A and A/D), modulators, instrumentation, etc. Extremely low-voltage supply and low-power integrated circuits are essential for battery-powered devices, portable and wearable biomedical electronics, IoT (Internet of Things) devices, and energy-autonomous integrated systems [1,2,3,4,5,6,7,8,9].

In the widely used bulk CMOS technology, several design techniques have been developed by designers to maintain acceptable circuit performance while the supply voltage (V_DD_) moves toward the threshold voltage (V_TH_) of the MOS transistor and even much lower, i.e., V_DD_ ≤ V_TH_ [1]. The most degraded parameter is the input voltage range, so techniques such as bulk-driven (BD), floating-gate (FG), quasi-floating gate (QFG), and multiple-input MOS transistors (MI-MOST) are good solutions to extend the input voltage range up to rail-to-rail at extremely low supply voltage [1,2,3,4,5,6,7,8,9].

OTA-C filters are widely used in signal processing applications and provide flexibility and electronic tunability compared to passive filters [10,11,12,13,14,15,16,17,18,19,20]. However, they have higher power consumption, and their structure is more complex. Therefore, special design considerations must be taken into account when using these filters for biosignal processing. Biosignals have the attributes of low amplitudes (µV–mV range) and low frequencies ≤10 kHz. Thus, a high dynamic range (DR) and a high linearity are required to enhance the signal-to-noise ratio and to minimize the harmonic distortion of filters, respectively. A fully differential structure of the filter is required to obtain improved signal voltage swing and noise immunity.

Therefore, in this paper, a multiple-input fully differential OTA-C filter for biosignal processing is presented. The multiple-input MOS transistor is used to reduce the design complexity and to keep the number of active blocks as small as possible. The circuit operates in the subthreshold regime to enable reduced voltage supply and power consumption of the proposed OTA. In addition, to avoid increasing the chip area and power consumption, a simple bulk-driven common-mode feedback circuit (CMFB) is incorporated into the OTA current mirror structure. All this makes the circuit simple and capable of operating from a low-voltage supply with low power consumption and extended input voltage range.

The paper is organized as follows: Section 2 describes the circuit of the proposed MI-FD OTA, and the application of the proposed OTA in a fifth-order Chebyshev low-pass filter, Section 3 presents the simulation results, and Section 4 concludes the paper.

## 2. Circuit Description

### 2.1. Multiple-Input Fully-Differential OTA

The circuit symbol of the multiple-input fully differential OTA (MI-FD OTA) is shown in Figure 1. Compared to the standard circuit (Figure 1a), the MI-OTA possess N differential inputs; thus, its output current is a superposition of responses from each differential input, as expressed by (1). In other words, the output current is a sum of N currents, which are products of differential voltages at each differential input and the same transconductance *g_m_*.
(1)$$I_{o}=\overset{N}{\underset{i=1}{\sum }}g_{m}\left(V_{+i}-V_{-i}\right)$$

The use of multiple-input OTAs increases the circuit versatility and reduces the number of active blocks used in some applications [8,9,10]. For instance, its application is especially advantageous in the design of OTA-C active filters.

The CMOS schematic of the multiple-input fully differential OTA proposed in this work is shown in Figure 2. In Figure 2a, a single-input (N = 1) core of the OTA is shown, while Figure 2b shows a complete version of the circuit with multiple inputs. Let us first consider the single-input core of Figure 2a. The proposed OTA exploits a classical current-mirror configuration, where the output currents of the differential stage (M_1_, M_2_, M_11_, M_12_) are transferred to the outputs O_+_ and O_−_ via the set of unity-gain current mirrors (M_3_–M_10_, M_13_, M_14_).

The input stage exploits the concept of a non-tailed differential pair [21]. The gates of the input transistors M_1_ and M_2_ are biased by the diode-connected transistors M_11_, M_12_. All transistors have their bulk terminals connected to the input terminals V_+_ and V_−_. Note that in this configuration, V_GS1_ = V_GS11_, V_BS1_ = V_BS11_, V_GS2_ = V_GS12_, and V_BS2_ = V_BS12_. Assuming perfect symmetry and that M_8_ = M_B_ and V_+_ = V_−_, the drain currents of M_1_ and M_2_ are equal to I_D1,2_ = (I_set_/2)[(W/L)_1,2_/(W/L)_11,12_]. This means that for common-mode signals (V_+_ = V_−_), while neglecting the second-order effects associated with the non-zero *g_ds_* conductances of MOS devices, the currents I_D1_ = I_D2_ remain constant, and the circuit is insensitive to such signals. On the other hand, for differential signals, the circuit behaves as a classical long-tailed bulk-driven (BD) differential pair [15]. In the weak inversion region, assuming I_D1_ = I_set_/2 at the operating point, its large-signal transfer characteristic can be expressed as follows [15]:(2)$$I_{D2}-I_{D1}=I_{set}\tanh\left(\frac{\eta \left(V_{+}-V_{-}\right)}{2n_{p}U_{T}}\right)$$
where *η* = *g_mb_*_1,2_/*g_m_*_1,2_ is the bulk to gate transconductance ratio of the input transistors M_1_ and M_2_ at the operating point, *n_p_* is the subthreshold slope factor for p-channel transistors and *U_T_* is the thermal potential. As can be concluded from (2), the linear range of the input pair is extended by a factor of 1/*η*, as compared to the classical GD differential pair.

The small-signal transconductance of the input stage, *g_mi_*, is equal to the bulk transconductance of the input transistors M_1_ and M_2_ and is given by:(3)$$g_{mi}=g_{mb1,2}=\eta \frac{I_{set}}{2n_{p}U_{T}}$$

Thus, the transistors M_11_ and M_12_ do not affect the input transconductance.

The unity-gain current mirrors of the OTA are based on self-cascode transistors M_i_-M_ic_ (i = 3–10, 13, 14). The transistors M_ic_ operate in the triode region, with a relatively small voltage drop V_ds_. Such a solution increases the output resistance of the OTA, and consequently its dc voltage gain, while not limiting its output voltage swing.

In order to stabilize the common-mode output level of this fully differential OTA, a simple BD common-mode feedback circuit (CMFB) is built into the structure of the p-channel current mirrors. Transistors M_10c_ and M_14c_ are split into two identical devices (c_1_ and c_2_), and their bulk terminals are connected to the outputs of the OTA. At the same time, the bulk terminals of M_9c_ and M_13c_ are connected to a reference voltage (GND in this case). When the output common-mode level of the OTA is increasing/decreasing with respect to GND, the drain currents of M_10_ and M_14_ are decreasing/increasing, thus compensating for the initial change of the common-mode level. Neglecting second-order effects, the CMFB does not affect the differential gain of the OTA. However, for large output amplitudes, such a simple CMFB introduces some distortion because variations of *r_dsc_*_1_ and *r_dsc_*_2_ are not symmetrical. Nevertheless, this effect is relatively weak for low V_DD_ (limited amplitude) and bulk-controlled transistors. Thanks to the proposed approach, we obtain a simple and sufficiently effective CMFB that does not consume additional power, does not increase the silicon area of the circuit, and, at the same time, does not limit the output voltage swing because it uses elements already existing in the circuit.

The output resistance of the OTA, determined for each single output, can be approximated by:(4)$$r_{out+/-}≅\left(g_{m8/5}r_{ds8/5}r_{ds8c/rc}\right)||\left(g_{m10/14}r_{ds10/14}r_{ds10c/14c}\right)$$
where *r_ds_*_10/14c_ = (*r_ds_*_10/14c1_)||(*r_ds_*_10/14c2_). Thus, the output resistance of the OTA based on self-cascode transistors is increased by a factor of *g_mi_r_dsci_*, where i denotes the index of the MOS transistors associated with the output nodes O_+_ and O_−_.

The dc differential voltage gain of the OTA in Figure 2a can be expressed as:(5)$$A_{vo}=2g_{mi}r_{out+/-}$$

Thanks to the increased value of the output resistance *r_out_*_+/−_, the value of A_vo_ is also increased, as opposed to the version with simple current mirrors.

The input-referred noise of the OTA is the same as for an OTA with the same topology, but with a BD differential pair, composed of transistors M_1_, M_2_ at the input, and biased with the same currents. Note that transistors M_11_ and M_12_ do not contribute to the output noise because their noise appears as a common-mode signal at the gates of M_1_ and M_2_.

Figure 2b shows the MI version of the proposed circuit. The multiple input is created by a capacitive divider composed of the capacitors C_Bi_. Their values should be significantly larger than the parasitic capacitances of the MOS transistors but much lower than the external capacitances used, for example, to provide a required frequency characteristic of a filter. In this way, the MI-OTA can be realized without the use of additional differential amplifiers, thus simplifying the overall structure and saving power. Moreover, the input signal range is extended since the input signal is attenuated by the capacitive divider. The input terminals of the internal OTA, i.e., the bulk terminals of M_1_ and M_2_, must be properly biased for dc, which is conducted by the MOS transistors M_b1_ and M_b2_. Since the transistors operate in a triode-cutoff region, with V_GS_ = 0, their *r_ds_* resistances are very high; thus, together with the capacitances C_Bi_, they create a high-pass (HP) filter with a very low cutoff frequency. Such a filter allows the dc component of a bio-signal to be removed as well. Further, at the operating point, V_DS11,12_ is approximately equal to zero, which means V_BS1,2_ and V_BS11,12_ are equal to zero as well. This lowers the bulk currents of the input transistors M_1,2_ and M_11,12_.

The transmittance of the input HP filter from a single input can be expressed as:(6)$$\beta \left(s\right)=\beta_{o}\frac{\frac{s}{\omega_{h}}}{1+\frac{s}{\omega_{h}}}$$
where *ω_h_* is the cutoff frequency of the filter, given as:(7)$$\beta \omega_{h}=\frac{1}{R_{LARGE}C_{\Sigma }}$$
where R_LARGE_ is the *r_ds_* resistance of M_b1_ and M_b2_, C_Σ_ = C_B1_ + C_B2_ + … + C_BN_ + C_i_ is the sum of all capacitances C_Bi_ of the input capacitive divider, and C_i_ is the input capacitance of the internal OTA, seen from the bulk of M_1_ (M_2_).

The high-frequency gain *β_o_* from the i-th input, with other inputs grounded, is given by:(8)$$\beta_{o}=\frac{C_{Bi}}{C_{\Sigma }}$$

For N = 3, and neglecting C_i_, *β_o_* = 1/3.

Taking into account the above considerations, the quasi-static transfer characteristic of the MI-OTA from the i-th input, with other inputs shorted to ground, and for frequencies higher than *ω_h_*, can be expressed as:(9)$$I_{O+}=-I_{O-}=I_{set}\tanh\left(\beta_{o}\eta \frac{\left(V_{i+}-Vi_{-}\right)}{2n_{p}U_{T}}\right)$$
thus, the linear range of the OTA is enlarged 1/*β_o_* times, as compared to the BD differential pair operating in a weak inversion, and 1/*β_o_η* times as compared to a GD pair. Assuming N = 3, and 0.18 μm technology, 1/*β_o_* = 3 and 1/*β_o_η* is typically around 10.

The small-signal transconductance of the MI-OTA from one differential input is consequently given by:(10)$$g_{mMI}=\beta_{o}\;\eta \frac{I_{set}}{2n_{p}U_{T}}$$
and is also lowered by the input capacitive divider, which also decreases the dc voltage gain.

The input capacitive divider also increases the input-referred noise of the MI OTA, which above the cutoff frequency *ω_h_* can be expressed as:(11)$$\bar{v_{nMI}^{2}}=\frac{1}{\beta_{o}^{2}}\bar{v_{n}^{2}}$$
where $\bar{v_{nMI}^{2}}$ is the input noise of the MI-OTA determined from one differential input and $\bar{v_{n}^{2}}$ is the input-referred noise of the single-input OTA of Figure 2a, which is equal to the input noise of the OTA with the same topology and biasing currents, but with the input stage based on a classical BD differential pair. Note that the transistors M_b1_ and M_b2_ (R_LARGE_) do not contribute to the total input noise since above *ω_h_*, their noise is suppressed by the capacitances C_Bi_.

As can be concluded from (11) and (9), due to the input capacitive divider, the input noise increases in the same proportion as the input linear range. This means that the dynamic range (DR) is the same with and without the input divider.

Concluding the above considerations, it is clear that the proposed MI-OTA shows increased versatility associated with the increased number of inputs. In addition, the input capacitive divider increases linear range, decreases the transconductance and dc voltage gain of the OTA, while its DR remains unchanged. At the same time, it performs the role of a HP filter, required in many biomedical applications to remove the dc component of the input signal.

### 2.2. Fifth-Order Chebyshev Low-Pass Filter

Figure 3 shows the fifth-order doubly terminated RLC ladder filter. The fifth-order Chebyshev low-pass prototype filter with 0.5 dB ripple was designed. Thus, the element values for the doubly terminated Chebyshev low-pass filter can be given as R_S_ = R_L_ = 1 Ω, L_1_ = L_2_ = 1.2296 H, C_1_ = C_5_ = 1.7058 F, and C_3_ = 2.5408 F. To design the 250 Hz cut-off frequency, the prototype element values of Figure 3 could be chosen as L_1_ = L_2_ = 54.74 kH, C_1_ = C_5_ = 15.529 pF, C_3_ = 23.131 pF, while letting R_S_ = R_L_ = 69.93 MΩ.

The proposed fifth-order Chebyshev low-pass filter using multiple-input OTAs is shown in Figure 4. By using multiple-input OTAs, five OTAs, and five capacitors are used. The resistors R_S_ and R_L_ can be implemented using the OTA, and their resistance values are given by R_S_ = $1/g_{m1}$ and R_L_ = $1/g_{m5}$, respectively. The floating inductors L_1_ and L_2_ are realized using gyrators and capacitors. Inductance values can be determined respectively by L_1_ = $C_{2}/g_{m2}^{2}$ and L_2_ = $C_{4}/g_{m4}^{2}$. The $g_{m1}-C_{1}$, $g_{m3}-C_{3}$, and $g_{m5}-C_{5}$ work as integrators. It should be noted that the structure of the filter is fully differential, which offers a high voltage swing compared with a single-ended filter, and the OTA-based circuit offers electronic tuning capability.

The proposed fifth-order Chebyshev low-pass filter in Figure 4 is designed to obtain the cut-off frequency of 250 Hz; thus, the active and passive values are given by $g_{m1-5}$ = 7.54 nS, C_1_ = C_5_ = 7.764 pF, C_2_ = C_4_ = 5.597 pF, and C_3_ = 11.565 pF.

## 3. Simulation Results

The MI-FD OTA and the filter application were designed and simulated in the Cadence environment using a 0.18 µm CMOS process from TSMC. A supply voltage of 0.5 V (V_DD_ = −V_SS_ = 0.25 V) was employed in the simulation, and the nominal static power consumption of the transconductor with I_set_ = 3 nA was 12 nW. The transistor aspect ratio is shown in Table 1.

The output current versus input voltage over the tuning range is shown in Figure 5. To obtain the dynamic characteristics of the BD-OTA and MIBD-OTA, a sine wave of amplitude 0.5 V at 200 Hz was applied to their inputs. Figure 5 shows the improved linearity of the MIBD-OTA compared with the BD-OTA with various setting currents I_set_ = (3, 6, 12, 24) nA.

Figure 6 shows the transconductance AC characteristics of the BD-OTA and MI-BD-OTA with different tuning currents I_set_ = (3, 6, 12, 24) nA. The transconductances were (−149, −144, −137.8, −132.2) dB and (−162.4, −156.7, −150.9, −139.6) dB, respectively. As can be seen, due to the high-pass filter created at the input of the MI-BD-OTA, the low cutoff frequency is around 1.5 Hz. This can be further reduced if needed by increasing the M_b_ resistance.

The equivalent input noise (µV/√Hz) of the (a) BD-OTA and (b) MI-BD-OTA is shown in Figure 7. The noise value was 2.51 µV and 11.5 µV at 100 Hz for the BD-OTA and MI-BD-OTA, respectively.

The AC transconductance characteristic from Figure 6b with I_set_ = 3 nA was replicated using 200 Monte-Carlo (MC) runs and process, voltage, and temperature (PVT) corners. The transconductance value at 100 Hz varied from −163.6 to −161.2 dB using MC. The MOS transistor process corners were fast-fast, fast-slow, slow-fast, and slow-slow. The MIM capacitor corners were fast-fast and slow-slow, and the transconductance varied from −162.9 to −161.9 dB. The low cutoff frequency varied from 0.5 to 6.7 Hz. For the voltage supply corners V_DD_ ± 10%V_DD_, the transconductance was −162.4 dB. The temperature corners were −10 °C and 60 °C, and the transconductance varied from −163 to −161.5 dB, while the low cutoff frequency varied from 0.16 to 11 Hz. These results are shown respectively in Figure 8a–d. All these variations are acceptable and expected, given the subthreshold operation. In addition, thanks to the circuit’s tunability, the desired transconductance value can be readjusted.

The power consumption of the filter for I_set_ = 3 nA was 5 × 12 nW = 60 nW. Figure 9a shows the frequency response of the LPF based on the RLC and the MI-BD-OTA with I_set_ = 3 nA. As can be seen, the two LP characteristics are close to each other, except for the low cut-off frequency at the sub-hertz level (50 mHz) created by the MI-BD-OTA input. This is an advantage in applications such as biosignals that require the removal of the DC component of the signal. The high cutoff frequencies are 242 Hz and 259 Hz, and the gain in the passband is −6 dB and −7.2 dB for the RLC and the filter based on MI-BD-OTA, respectively. Figure 9b shows the tuning capability of the filter with different I_set_ = (3, 6, 12, 24) nA. The high cutoff frequency was (259, 5094, 985, 1880) Hz, while the low cutoff frequency was around 50 mHz.

Figure 10a shows the transient response of the LPF, where the input signal was a sinewave with a rail-to-rail amplitude at 50 Hz. The spectrum of the output voltage is shown in Figure 10b and shows 0.97% total harmonic distortion (THD). Hence, the input signal can be rail-to-rail with less than 1% THD.

Figure 11 shows the total harmonic distortion of the output signal for different amplitudes of the input sine signals at 50 Hz. With an input amplitude of 400 mV and less, the THD was around 0.4%. The circuit can even operate beyond the rail-to-rail level with 3% THD for 600 mV amplitude.

Figure 12 shows the equivalent output noise of the LPF. The integrated output noise in the range of 50 mHz to 259 Hz band was calculated to be 772 µV, giving a dynamic range of 53.2 dB.

To demonstrate the filtering function of the LPF with an ECG signal, Figure 13a shows the noisy input signal obtained by a clean ECG signal combined with 3 mV signals at 10 mHz and 500 Hz. Figure 13b shows the output signal of the filter where the low and high frequency signals have been filtered out.

The proposed low-pass filter was compared with some previous works, as shown in Table 2. The fifth-order filters in [8,9,19,20,22,23] have been selected for comparison. It is clear that the MI-OTA-based filter offers a minimum number of active OTAs with a fully differential structure compared with [19,20,22,23]. Compared with [19,22,23], the proposed filter has low power consumption compared with [19,20,23], the proposed filter obtains low voltage capability, and compared with [8,19,20,22,23], the proposed filter provides LPF/BPF. Finally, compared with the BPF in [9], the proposed filter provides fully differential voltages, higher order (5 versus 3), with fewer active and passive components (5 versus 6). This is due to the internal high-pass filter created at the input of the MI-BD-OTA.

## 4. Conclusions

This paper presents a 0.5 V multiple-input fully differential OTA and its application to a fifth-order Chebyshev low-pass filter for bio-signal processing. The circuit uses several design techniques and operates in the subthreshold region with a 0.5 V supply. It consumes power in the range of a few nanowatts while offering an extended input voltage range. The fifth-order Chebyshev low-pass filter, which uses the proposed multiple-input fully differential OTA as an active block, offers a minimum number of active devices and low design complexity. The circuit performance and applications have been confirmed by intensive simulation in the Cadence program.

## Figures and Tables

**Figure 1 sensors-24-02150-f001:**
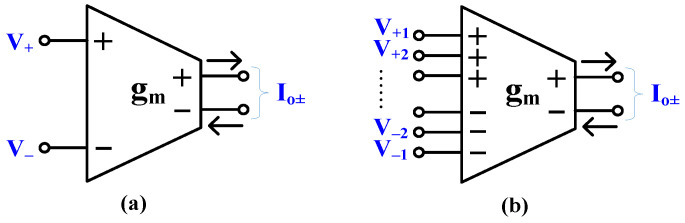
Electrical symbol of (**a**) fully differential OTA and (**b**) multiple-input fully differential.

**Figure 2 sensors-24-02150-f002:**
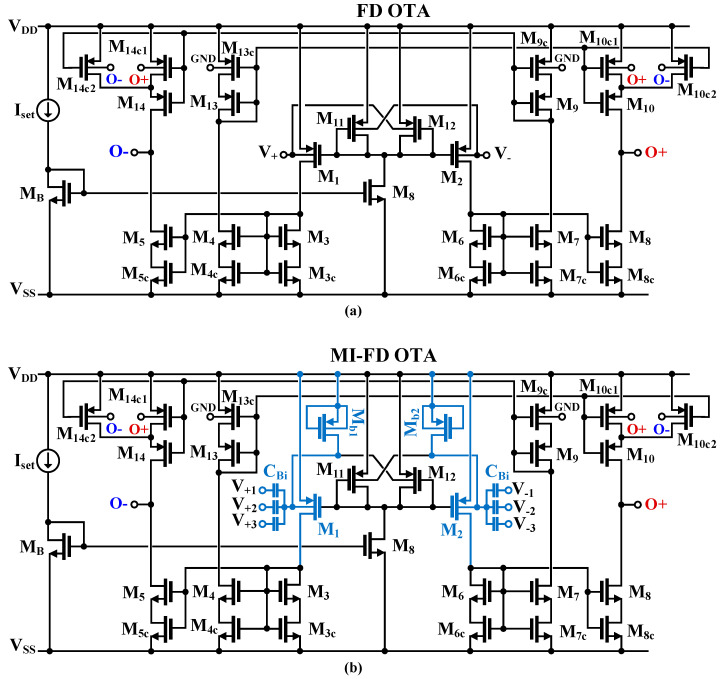
CMOS implementation of (**a**) fully differential OTA and (**b**) multiple-input fully differential OTA.

**Figure 3 sensors-24-02150-f003:**
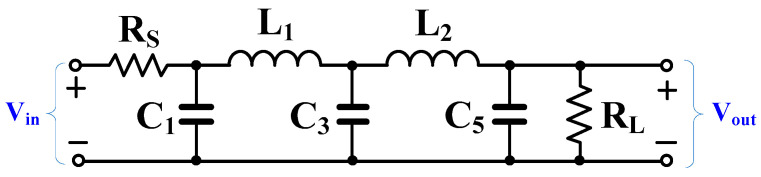
Fifth-order doubly terminated RLC ladder filter.

**Figure 4 sensors-24-02150-f004:**
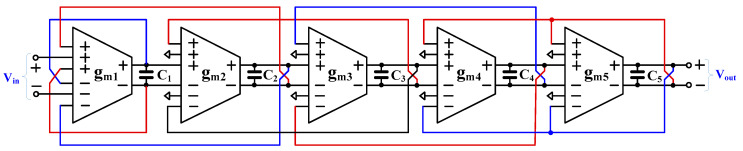
Proposed fifth-order Chebyshev low-pass filter.

**Figure 5 sensors-24-02150-f005:**
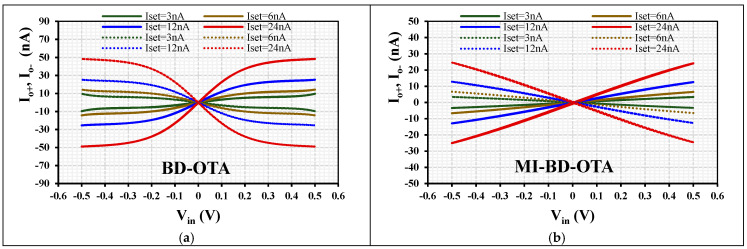
The output current I_o+_ and I_o−_ (dashed line) versus input voltage V_in_ over the tuning range of the (**a**) BD-OTA and (**b**) MIBD-OTA with different setting currents.

**Figure 6 sensors-24-02150-f006:**
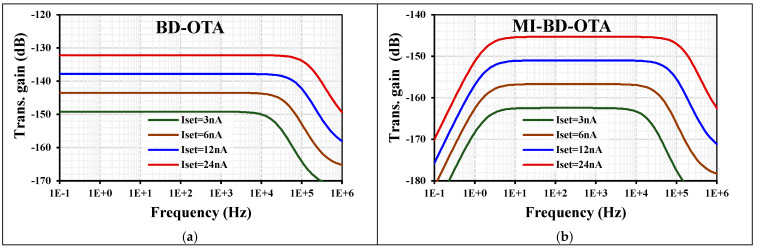
The transconductance AC characteristic of the (**a**) BD-OTA and (**b**) MIBD-OTA with different setting currents.

**Figure 7 sensors-24-02150-f007:**
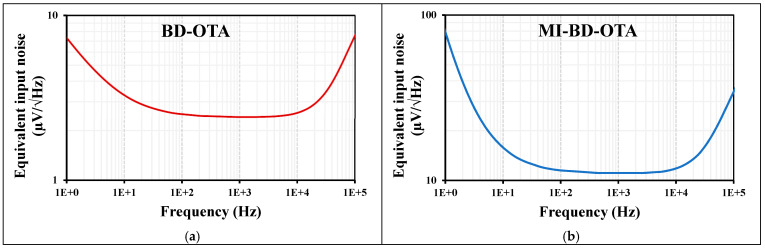
The equivalent input noise (µV/√Hz) of the (**a**) BD-OTA and (**b**) MIBD-OTA.

**Figure 8 sensors-24-02150-f008:**
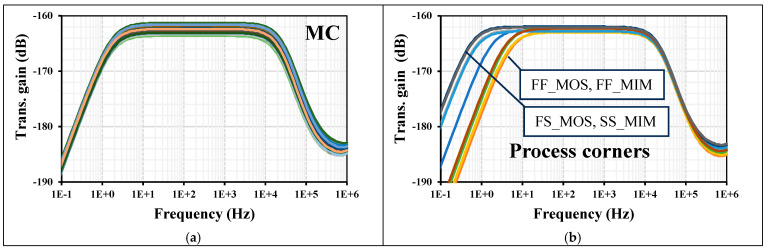
The transconductance AC characteristic of the MI-BD-OTA: (**a**) MC, (**b**) process, (**c**) voltage, and (**d**) temp. corners.

**Figure 9 sensors-24-02150-f009:**
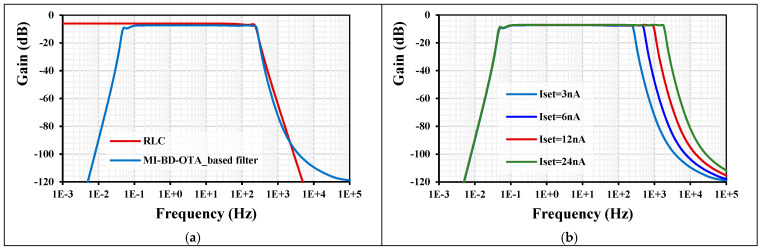
(**a**) The frequency response of the LPF and (**b**) the tuning capability of the LPF.

**Figure 10 sensors-24-02150-f010:**
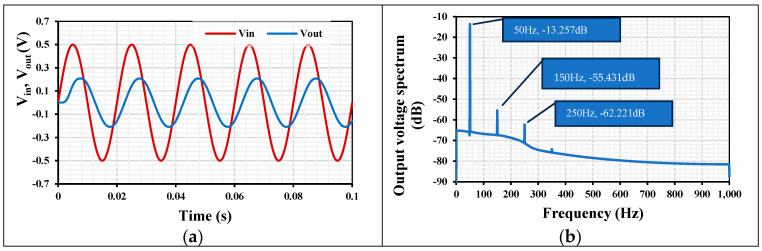
(**a**) The transient response of the LPF and (**b**) the output spectrum.

**Figure 11 sensors-24-02150-f011:**
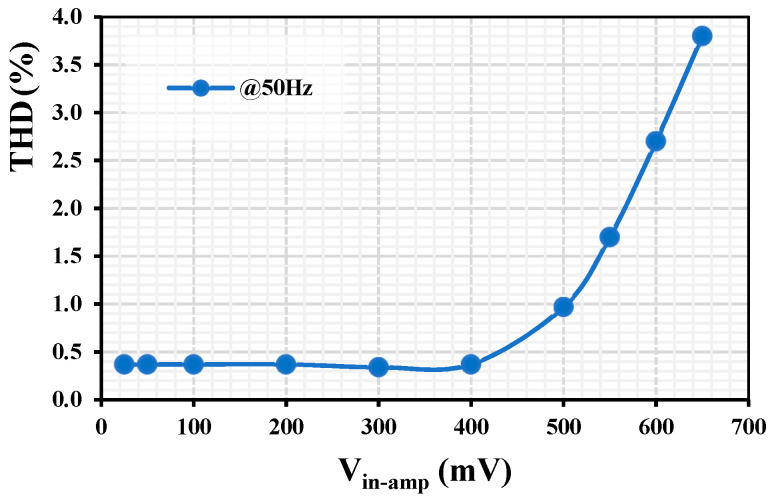
The THD of the LPF with different input amplitudes.

**Figure 12 sensors-24-02150-f012:**
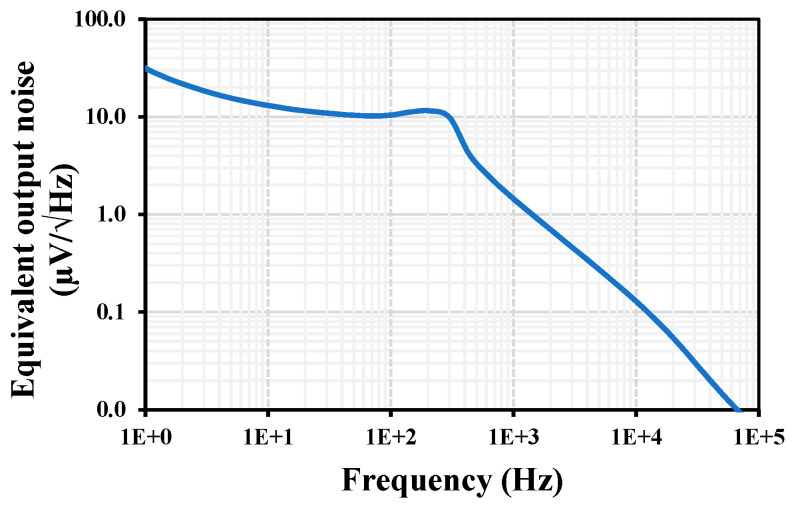
The equivalent output noise of the LPF.

**Figure 13 sensors-24-02150-f013:**
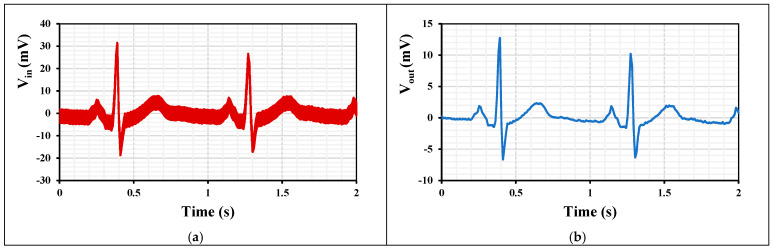
The transient response of the LPF to an ECG signal: (**a**) noisy input signal and (**b**) filtered output signal.

**Table 1 sensors-24-02150-t001:** Component parameters of the BD-OTA and MI-BD-OTA.

Transistor	W/L (µm/µm)
M_1_, M_2_	2 × 20/3
M_11_, M_12_	20/3
M_b_, M_8_	5/1
M_3_–M_8_, M_9_–M_12_	20/1
M_3c_–M_8c_	20/2
M_9c_, M_11c_	2 × 10/3
M_10c1_, M_10c2_, M_11c1_, M_11c2_	10/3
M_b1_, M_b2_	5/6
C_B_ = 0.5 pF	

**Table 2 sensors-24-02150-t002:** Performance comparison of the proposed filter and previous fifth-order low-pass filters.

Symbol	This Work	Sensors (2020) [8]	IEEE ACCESS (2022) [9]	IEEE TCAS-II (2018) [19]	IEEE TBioCAS (2019) [20]	MEJ (2019) [22]	IEEE ACCESS (2023) [23]
V_DD_ [V]	0.5	0.5	0.5	1.0	1.0	0.25	1.0
Tech [μm]	0.18	0.18	0.18	0.18	0.18	0.13	0.065
V_TH_ [V]	0.5	0.5	0.5	0.5	0.5	0.44	0.3
Order (N)	5	5	3	5	5	5	5
Filter type	Chebyshev	Butterworth	Chebyshev	Butterworth	Butterworth	Butterworth	Butterworth
Filtering function	LPF/BPF	LPF	BPF	LPF	LPF	LPF	LPF
No. of active device	5 MI-OTAs	5-MI-OTAs	6-MI-OTAs	6 OTAs	6 OTAs	6 FDDTAs	11 OTAs
Architecture	G_m_-C fully diff.	G_m_-C fully diff.	G_m_-C single-end	G_m_-C fully diff.	G_m_-C fully diff.	G_m_-C fully diff.	G_m_-C fully diff.
MOST technique	MIBD	MIGD	MIBD	GD	GD	GD	GD
BW/central freq. [Hz]	250	250	250	250	250	100	1 × 10^6^
DR [dB]	53.2	63.24	60.4	49.9	61.2	57	50.54
Power (P) [nW]	60	34.65	60	350	41	603	167 × 10^3^
FOM = P/(N × BW × DR) [pJ]	0.9	0.43	1.32	0.896	0.0286	1.7	0.66
LV operation capability = V_TH_/V_DD_ × 100 [%]	100	100	100	50	50	176	30
Area [mm^2^]	NA	NA	0.036 (off chip cap.)	0.12	0.14	0.67 (off-chip cap.)	0.0164
Obtained results	Simulated	Simulated	Post-layout	Measured	Measured	Measured	Measured

Note: MIBD = Multiple-Input Bulk-Driven, GD = Gate-Driven, MIGD = Multiple-Input Gate-Driven, LPF = Low-Pass Filter, BPF = Band-Pass Filter.

## Data Availability

Data are contained within the article.

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
