# Peer review of "0.5 V Multiple-Input Fully Differential Operational Transconductance Amplifier and Its Application to a Fifth-Order Chebyshev Low-Pass Filter for Bio-Signal Processing"

_sensors, 2024, doi:10.3390/s24072150_

Round 1

Reviewer 1 Report

Comments and Suggestions for Authors

Thank you so much for the well-written and well-structured paper. However, please clarify the following queries.

Please place all the symbols or parameters notation with proper subscripts and superscripts. It can be observed that in most of the Figs, the subscripts are not properly placed.

Authors are advised to put the axis parameters in terms of numeric rather than 1E+0 format.

Please discuss “Table 2. Performance comparison of the proposed filter and previous fifth-order low-pass filters.” In detail, also highlight the obtained data are better in terms of what.

Please discuss the key considerations and limitations of the Monte Carlo analysis.

Authors are advised to incorporate some of the latest quality references.

Comments on the Quality of English Language

Minor editing of the English language is required

Author Response

Reviewer 1

Comments and Suggestions for Authors

Thank you so much for the well-written and well-structured paper. However, please clarify the following queries.

Please place all the symbols or parameters notation with proper subscripts and superscripts. It can be observed that in most of the Figs, the subscripts are not properly placed.

Response

Thank you very for positive feedback.

Text in Figures such as Figs. 5-6 is the format of Microsoft Excel that we cannot edit to the subscript.

Authors are advised to put the axis parameters in terms of numeric rather than 1E+0 format.

Response

Unfurtantly using numeric format the numbers are overlapped that why we keep the current format.

Please discuss “Table 2. Performance comparison of the proposed filter and previous fifth-order low-pass filters.” In detail, also highlight the obtained data are better in terms of what.

Response

The performance of the circuit is explained, and the text is highlighted, e.g.:

compared with [8, 19-22], the proposed filter provides LPF/BPF. Finally compared with the BPF in [9], the proposed filter provides fully differential voltages, higher order (5 versus 3) with fewer number of active and passive components (5 versus 6). This is due to the internal high-pass filter created at the input of the MI-BD-OTA.

Please discuss the key considerations and limitations of the Monte Carlo analysis.

Response

Additional text has been added.

All these variations are acceptable and expected given the subthreshold operation. In addition thanks to the circuit's tunability, the desired transconductance value can be readjusted.

Authors are advised to incorporate some of the latest quality references.

Response

References has been updated.

Reviewer 2 Report

Comments and Suggestions for Authors

The paper discusses the design of a multiple input bulk-driven fully differential operational transconductance amplifier (MI-FD OTA) that is suitable for low voltage and low power applications. To meet these requirements, the transistors have been designed to work in the subthreshold region. The authors have also designed a fifth-order Chebyshev low-pass filter for biomedical applications using the proposed MI-FD OTA to prove the validity of the concept.

The brief introduction discusses the existing solutions in literature for reducing the supply voltage towards or even below the threshold voltage, and their applications in OTA filters. 

  • A more detailed presentation of the issues that emerge from the presented solutions that would better highlight the advantages of the solution adopted for the design would be welcome.

The next section of the work is dedicated to a detailed presentation of the proposed solution for the MI-FD OTA and its application in a fifth-order low-pass filter. 

  • Aren't the component values on page 6, lines 199-200, the normalized component values from standard design tables for the desired prototype, and the values from lines 201-202 the values obtained by frequency (250Hz) and impedance (69.93 MΩ) scaling? If so, then you don't have units of measure.

To demonstrate the capabilities of the proposed solution, the following chapter presents some results obtained from analyses performed in the Cadence environment. 

  • Can you please clarify if the aspect ratios of the transistors listed in Table 1 were chosen to ensure their correct operation in the subthreshold region when subjected to the maximum current used in the analyses? The statement in line 219 suggests that the design was intended for a current of 3nA, but some analyses indicate a current of 24nA.
  • What is the calculation formula for the power value from line 219? who is 8 in the formula? 
  • It is more appropriate to rename the Oy axis on Figure 6 and Figure 8 to OTA gain since dB is used as a unit of measure for gain and not transconductance.

The conclusions section should be expanded, possibly through a more detailed description of the results obtained.

Author Response

Reviewer 2

Comments and Suggestions for Authors

The paper discusses the design of a multiple input bulk-driven fully differential operational transconductance amplifier (MI-FD OTA) that is suitable for low voltage and low power applications. To meet these requirements, the transistors have been designed to work in the subthreshold region. The authors have also designed a fifth-order Chebyshev low-pass filter for biomedical applications using the proposed MI-FD OTA to prove the validity of the concept.

The brief introduction discusses the existing solutions in literature for reducing the supply voltage towards or even below the threshold voltage, and their applications in OTA filters.

  • A more detailed presentation of the issues that emerge from the presented solutions that would better highlight the advantages of the solution adopted for the design would be welcome.

Response:

Additional text has been updated.

The next section of the work is dedicated to a detailed presentation of the proposed solution for the MI-FD OTA and its application in a fifth-order low-pass filter.

  • Aren't the component values on page 6, lines 199-200, the normalized component values from standard design tables for the desired prototype, and the values from lines 201-202 the values obtained by frequency (250Hz) and impedance (69.93 MΩ) scaling? If so, then you don't have units of measure.

Response:

If we need the cut-off frequency 250Hz, scaling is need and R=1 become 69.9M (1/gm).

To demonstrate the capabilities of the proposed solution, the following chapter presents some results obtained from analyses performed in the Cadence environment.

  • Can you please clarify if the aspect ratios of the transistors listed in Table 1 were chosen to ensure their correct operation in the subthreshold region when subjected to the maximum current used in the analyses? The statement in line 219 suggests that the design was intended for a current of 3nA, but some analyses indicate a current of 24nA.

Response

The circuit was designed to obtain quiescent voltages at all circuit nodes of VDD/2, that means VGS of all transistors equal to VDD/2 as well, for a nominal biasing current of 3nA. This allows maximum available variations of VGS under PVT, signal swing and changes of the biasing current, that maximizes the possible tunning range of the circuit. Note, that changing the biasing current 8 times with respect to its nominal value, changes the VGS value of all transistors by about 75mV only, still  leaving sufficient room for PVT variations and signal swing (VGS can change from VDSsat to VDD-VDSsat, i.e. from 100 to 400mV).

  • What is the calculation formula for the power value from line 219? who is 8 in the formula?

Response

Power= Itotal*VDD= (8 branches * 3uA)*0.5. However, we have kept only the value of the power consumption to avoid confusing.

  • It is more appropriate to rename the Oy axis on Figure 6 and Figure 8 to OTA gain since dB is used as a unit of measure for gain and not transconductance.

Response

These figures have been changed.

The conclusions section should be expanded, possibly through a more detailed description of the results obtained.

Response

The text is highlighted.
